# Neuroimaging Methods to Map In Vivo Changes of OXPHOS and Oxidative Stress in Neurodegenerative Disorders

**DOI:** 10.3390/ijms23137263

**Published:** 2022-06-30

**Authors:** Jannik Prasuhn, Liesa Kunert, Norbert Brüggemann

**Affiliations:** 1Institute of Neurogenetics, University of Lübeck, 23538 Lübeck, Germany; jannik.prasuhn@neuro.uni-luebeck.de (J.P.); liesa.kunert@student.uni-luebeck.de (L.K.); 2Department of Neurology, University Medical Center Schleswig Holstein, Campus Lübeck, 23538 Lübeck, Germany; 3Center for Brain, Behavior and Metabolism, University of Lübeck, 23562 Lübeck, Germany

**Keywords:** neuroimaging, mitochondria, mitochondrial dysfunction, neurodegeneration, Parkinson’s disease (PD)

## Abstract

Mitochondrial dysfunction is a pathophysiological hallmark of most neurodegenerative diseases. Several clinical trials targeting mitochondrial dysfunction have been performed with conflicting results. Reliable biomarkers of mitochondrial dysfunction in vivo are thus needed to optimize future clinical trial designs. This narrative review highlights various neuroimaging methods to probe mitochondrial dysfunction. We provide a general overview of the current biological understanding of mitochondrial dysfunction in degenerative brain disorders and how distinct neuroimaging methods can be employed to map disease-related changes. The reviewed methodological spectrum includes positron emission tomography, magnetic resonance, magnetic resonance spectroscopy, and near-infrared spectroscopy imaging, and how these methods can be applied to study alterations in oxidative phosphorylation and oxidative stress. We highlight the advantages and shortcomings of the different neuroimaging methods and discuss the necessary steps to use these for future research. This review stresses the importance of neuroimaging methods to gain deepened insights into mitochondrial dysfunction in vivo, its role as a critical disease mechanism in neurodegenerative diseases, the applicability for patient stratification in interventional trials, and the quantification of individual treatment responses. The in vivo assessment of mitochondrial dysfunction is a crucial prerequisite for providing individualized treatments for neurodegenerative disorders.

## 1. Introduction

### 1.1. What Is Mitochondrial Impairment? The Molecular Complexity of a Fundamental Cell Organelle

Bioenergetic disturbances in the nervous system have been identified as a pathophysiological hallmark in many neurodegenerative diseases (NDs), including idiopathic Parkinson’s disease (PD) [1], atypical Parkinson’s disease (APD) [2], Alzheimer’s disease (AD) [3], among other forms of dementia, Huntington’s disease (HD) [4], prion disease [5], motor neuron disease (MND) [6], and certain forms of ataxia [7]. Most metabolic pathways (e.g., the tricarboxylic acid cycle, TCA) converge to the mitochondria and thus have an impact on the final steps of energy production by oxidative phosphorylation (OXPHOS) [1]. Yet, considering mitochondrial dysfunction purely by the resulting bioenergetic disturbances would be an oversimplification [1]. As a fundamental cellular organelle, mitochondria have widespread interconnections to the overall cellular homeostasis, making elucidating precise disease mechanisms of NDs challenging [8]. In addition, our current understanding of the molecular mechanisms underlying most NDs is incomplete and likely involves many pathophysiological events and processes [9]. Whether mitochondrial dysfunction can be considered the primary driver or simply a bystander of neurodegeneration largely remains to be elucidated [10]. Besides the overall complexity of disease mechanisms in NDs, *mitochondrial dysfunction* can refer to various unphysiological molecular processes [1]. These processes extend but are not limited to impaired mitochondrial biogenesis, dynamics and trafficking, calcium and metal ion dyshomeostasis, heme biosynthesis, control of cell division and cell fate decisions, and neuroinflammation [1]. The many-faceted dysfunction of mitochondria ultimately results in impaired energy supply and increased oxidative stress—both mechanisms leading to neurodegeneration, disease manifestation, and progression [8]. Therefore, these mechanisms could serve as viable surrogate markers for mitochondrial dyshomeostasis and can be mapped by innovative neuroimaging methods. 

### 1.2. Why Could It Be Helpful to Identify Patients with Predominant Mitochondrial Dysfunction? On the Leap to Individualized Treatment Decisions

NDs are debilitating diseases, substantially impacting the well-being of patients and caregivers. The rising prevalence provides a significant socioeconomic burden to aging populations [11] and cannot be solely explained by the shifting age structure in modern societies [12]. Only symptomatic treatments are available to date, and no drug has been shown to reveal any clinically relevant disease-modifying properties [13]. NDs follow an individual time course often preceded by a prodromal phase [14]. Substantial neuronal cell loss has already occurred in prodromal phases, not yet having reached a symptom-causing threshold to facilitate a clinically established diagnosis. However, it would be highly desirable to identify patients in the prodromal phase to start neuroprotective treatments before disabling symptoms have occurred [14]. For some NDs, this prodromal phase can be outlined by clinical criteria (e.g., for PD) [15] or by pre-symptomatic genetic testing (e.g., for HD) [16]. Nevertheless, identifying study participants in a prodromal phase causes ethical dilemmas in conceptualizing clinical trials. Clinical concepts of the prodromal phase also imply that not all identified individuals will manifest their suspected disease [14]. Drug candidates must meet exceptionally high safety standards to be considered for clinical testing in pre-diseased individuals. Accompanying diagnostics could substantially enrich the frequency of disease-converters to ensure clinical trial success [17]. In NDs, the symptom severity and disease progression can be highly individual [18,19,20]. In addition, clinical trials in NDs evaluate disease-modifying properties of candidate drugs following long interventional periods [14]. Pathophysiology-orientated biomarkers and adaptive clinical trial designs will substantially improve the efficiency of neuroprotective trials by identifying surrogate markers for treatment response. These considerations could enrich study participants suitable for targeted therapies in innovative clinical trial designs. Patients suffering from NDs with suspected mitochondrial dysfunction are likely the most promising candidates for such innovative clinical trials. Human metabolism is a highly dynamic system. Therefore, immediate treatment responses following mitochondria-targeted therapies (e.g., by the pharmacological enhancement of OXPHOS) could be dynamically mapped within the scope of adaptive trials. Subsequently, reliable data can be generated to test if the paraclinical improvement of mitochondrial dysfunction can result in clinically relevant disease modification. Even though our current understanding of disease biology is constantly expanding, the molecular events leading to NDs are complex. They involve an interplay of various molecular events within an individual patient and disease course [21]. The molecular heterogeneity of NDs makes it unlikely that a single drug target can recapitulate all pathophysiological hallmarks of a given disease entity at any given time. A combination of potentially disease-modifying treatment strategies will likely be combined in a tailored and highly individualized fashion [21]. Developing reliable and pathophysiology-orientated biomarkers is a compulsory prerequisite for individualized treatment regimes. However, the role of mitochondrial dysfunction in the development and progression of NDs is undisputed [22]. Genetic insights and pathway-based analyses help elucidate mitochondrial dysfunction’s complexity in NDs. We refer the reader to a review article illustrating how these approaches help identify potential treatment strategies in patients with genetic and non-genetic Parkinson’s disease as a neurodegenerative model disease [23]. Although the evolvement and the individual time course of suspected mitochondrial dysfunction in NDs are only poorly understood, mitochondria-targeted therapies can still be considered a viable treatment strategy for many of these disorders [8]. The convergence of other disease mechanisms on mitochondrial homeostasis may help to reduce the complexity of drug development and testing [24]. The unknown temporal dynamics of mitochondrial dysfunction may result in a *cause or consequence dilemma* for our current understanding of disease biology [25]. The vital role of mitochondrial homeostasis for neuronal survival does not necessarily require causality in identifying mitochondria as viable treatment targets. For example, repeat expansions in the *HTT* gene cause HD mainly by protein misfolding and the toxic aggregation properties of the mutated HTT protein [26]. Even though mitochondrial dysfunction may not be the primary cause of neurodegeneration, it can be considered a direct consequence thereof [4]. Restoring mitochondrial homeostasis in HD could improve neuronal survival in HD. The clinical and molecular intricacy of NDs results in a pressing need for established biomarkers of mitochondrial dysfunction [27]. These biomarkers will shed light on the unclear clinical and molecular temporal dynamics of disease onset and progression and improve our current understanding of relevant disease mechanisms and the efficacy of clinical trials by dynamically monitoring responses to mitochondria-targeted treatments. Some promising methods are already available to establish potential biomarkers of mitochondrial dysfunction in vivo.

### 1.3. The Unbundling of Metabolic Pathways: Mitochondria at the Convergence of Human Metabolism

Mammalian metabolism is highly interwoven. Most nutrients are broken down into intermediary metabolites of the TCA. They can enter OXPHOS via complex I (by NADH) or complex II (by FADH_2_) of the electron transport chain (ETC) [8]. The metabolic conflux to the ETC offers possibilities to measure mitochondrial impairment indirectly. Upstream metabolites (e.g., of the TCA cycle) can subsequently accumulate to impaired OXPHOS (see Figure 1).

Moreover, neurons may switch to non-OXPHOS catabolism to generate energy (e.g., by anaerobic glycolysis) [28]. In the past, in vivo measurements of lactate (as the end route of anaerobic glycolysis) have been evaluated to map mitochondrial impairment in patients with NDs [29] (see Figure 2). 

Interestingly, other catabolic pathways (e.g., fatty acid breakdown by β-oxidation) are located in the mitochondria and can also be considered disease-relevant surrogate markers of mitochondrial dysfunction [30]. The interwovenness of mammalian metabolism also impacts the specificity of single molecules to quantify mitochondrial impairment in patients with NDs. In redox reactions, substrates of ETC complexes are often involved in various cellular pathways (e.g., NADH/ NAD^+^ as a ubiquitous coenzyme in redox reactions) [31]. The respective lack of pathophysiological specificity must be considered critically in conceptualizing studies probing mitochondrial impairment in NDs. However, these limitations extend to all studies on human metabolism and are not intrinsic to neuroimaging methods. Subsequently, quantifying single metabolites can only be interpreted as an estimation of in vivo mitochondrial impairment. 

### 1.4. Neuroimaging for Patient Stratification and Therapy Monitoring in Patients with Suspected Mitochondrial Dysfunction?

The human brain accounts for only ~2% of the total body weight [32]. Based on the striking biomass difference between the human brain and the overall organism, it appears doubtful whether biomarkers derived from peripheral tissue (e.g., blood) can yield substantial diagnostic value in identifying NDs patients with cerebral mitochondrial dysfunction [1]. This is particularly important if only distinct neuronal subpopulations (e.g., hippocampal neurons in AD) are predominantly involved in disease development [33]. In marked contrast, the brain requires ~20% of the total cardiac output volume and ~25% of the human overall energy expenditure, pointing towards an exceptionally high metabolic demand [32]. Most of the organism’s metabolic activity is forwarded to the generation of ATP (by OXPHOS). Neuroimaging methods addressing the cerebral energy metabolism could therefore result in a high diagnostic yield to non-invasively identify patients with suspected mitochondrial dysfunction.

Interestingly, the study of brain energy metabolism has been one of the key interests of neuroscientists since the earliest neuroimaging methods have been available. For example, the biological origin of the blood-oxygen level-dependent imaging contrast (BOLD; frequently applied in functional MRI studies) has been elucidated for decades [34]. However, it took a substantial amount of time to advance this method to study in vivo oxygen consumption rates [35]. There seems to be a missing link between the exciting and rapidly expanding field of novel neuroimaging methods and the necessary translation into clinical practice. Indeed, there are significant challenges related to the clinical applicability of many of the following methods. We will discuss shortcomings and advancements required to facilitate the future applicability of these methods in clinical settings.

### 1.5. What Can We Measure? Mitochondrial Bioenergetics and Oxidative Stress as Promising Neuroimaging Markers of Mitochondrial Dysfunction

The synthesis of ATP by OXPHOS is recognized as the most prominent function of mitochondria. During OXPHOS, electrons can leak from the ETC and react with O_2_ to, e.g., form superoxides (O_2_^−^, one class of reactive oxygen species, ROS) [36]. Electron leakage constantly occurs during physiological conditions, and the resulting ROS are removed by various antioxidant-active coping mechanisms [36]. Mitochondrial dysfunction can negatively affect this process and severely disturb physiological OXPHOS [8]. The harmful elevation of ROS has several negative downstream effects, e.g., damaging proteins, which alter cellular homeostasis and impair neuronal survival [37]. Our current understanding of NDs suggests that many pathophysiological hallmarks directly cause or result from OXPHOS disturbances and oxidative stress [8]. Therefore, the in vivo assessment of OXPHOS and oxidative stress appear viable targets for neuroimaging methods (see Figure 1). Besides OXPHOS and oxidative stress, other relevant pathophysiological aspects of NDs linked to mitochondrial dysfunction could be assessed by neuroimaging methods. Three prominent examples are:the investigation of OXPHOS-upstream metabolic pathways (e.g., the TCA cycle) or the non-OXPHOS generation of ATP (e.g., by anaerobic glycolysis) [38,39],the determination of brain iron deposition by iron-sensitive magnetic resonance imaging (MRI) contrasts [40,41], andthe assessment of neuroinflammatory surrogate markers (e.g., by fluid-sensitive MRI sequences or ligand-specific positron emission tomography (PET) imaging) [42,43,44].

This review will focus on neuroimaging methods to map in vivo changes of OXPHOS and oxidative stress. These methods are based on various physical phenomena to generate molecule-specific image contrasts. Here, we distinguish three different physical phenomena these methods are based on: Nuclear magnetic resonance (NMR). Atomic nuclei with a non-zero nuclear spin can be considered NMR-active. MRI and magnetic resonance spectroscopy (MRSI) utilize this phenomenon to generate image contrast or simultaneously measure multiple metabolites. In most MRI or MRSI studies, the NMR-phenomenon of protons (^1^H) is used to derive imaging data. However, these methods can also be applied to other NMR-active nuclei (e.g., ^13^C-, ^15^N-, ^17^O-, or ^31^P-nuclei). The use of non-^1^H nuclei in neuroimaging is often referred to as hetero-, multi-, or X-nuclear MR(S)I [38,45]. The acquisition of adequate X-nuclear MRSI signals critically depends on the natural abundance, the relative sensitivity (defined by the gyromagnetic ratio and the nuclear spin), and T1/T2 relaxation times (using quadrupolar relaxation) of the NMR-active isotope [38,46].Near-infrared spectroscopy (NIRS). NIRS is a physical analysis technique based on the absorption and emission of short-waved (within the near-infrared region of the electromagnetic spectrum) light to detect chromophoric metabolites (e.g., hemoglobin) [47].Radioactive decay. PET imaging is based on the simultaneous detection of two gamma-ray photons produced after a positron-emitting radionuclide (β+ decay) decay. The target-specific generation of radiotracers and the distribution of these weakly radioactively labeled substances can be used to image biochemical and physiological functions of the brain [48].

### 1.6. The Scope of this Review

In this review, we will summarize how distinct neuroimaging methods can be employed to probe mitochondrial dysfunction in vivo and why this could be relevant for improving NDs patient care. We will focus on neuroimaging methods that can directly or indirectly map OXPHOS impairment and oxidative stress, both viable surrogate markers of mitochondrial dysfunction. In addition, we will discuss which steps are needed to translate recent methodological advancements into clinical practice. 

## 2. Neuroimaging-Based Assessments of OXPHOS-Related Complexes and Metabolites

In post-mortem studies, a significant dysregulation of ETC complexes was observed in patients with NDs [49]. These findings strongly implicate that mitochondrial dysfunction-linked alterations in OXPHOS can be considered a highly relevant molecular mechanism in different NDs. Histopathological examinations revealed decreased complex I level, preferentially in the substantia nigra (SN), in patients suffering from PD [50]. These findings are consistent with the fact that inhibitors of complex I (such as the environmental toxins MPTP or rotenone) can cause parkinsonism in animal models and humans [51]. HD has been associated with defects of complex II and, to a lesser extent, complex IV [52]. The chronic administration of the complex II inhibitor 3-nitropropionic acid causes an HD-like phenotype in rodent and non-human primate models [53]. In AD, widespread cortical complex IV defects were identified in post-mortem brain tissue [54]. The in vivo neuroimaging-based assessment of ETC-related metabolite levels could thus help elucidate the complex role of OXPHOS disturbances in NDs.

### 2.1. ^31^Phosphorus-Magnetic Resonance Spectroscopy Imaging to Quantify OXPHOS-Related Metabolite Levels In Vivo

In general, MRSI can provide a unique view into these metabolic processes in vivo [38]. NMR-active nuclei are partially shielded from the static magnetic field by surrounding electrons, which causes slight magnetic field distortions. These distortions result in so-called chemical shifts (usually expressed in parts per million, ppm), which can be used to identify distinct metabolites in a spectrum based on their known molecular structure [55]. Besides the flexibility and widespread applicability of ^1^H-MRSI, ^31^P-MRSI offers unique opportunities to investigate OXPHOS-related mitochondrial dysfunction in vivo. The ^31^P nucleus has a significantly lower gyromagnetic ratio compared to ^1^H. However, the natural abundance of the ^31^P nucleus in living tissues is close to 100%, resulting in robust NMR signals [56]. Therefore, ^31^P-MRSI can be applied to map high-energy phosphorus-containing metabolites (HEPs), such as ATP or phosphocreatine (PCr), in vivo [56] (see Figure 3).

HEPs can be interpreted as the end route of OXPHOS. In previous studies, HEPs have been expressed as a ratio to inorganic phosphate (iP) to account for intraindividual differences, e.g., in the alimentary intake of phosphorus-containing nutrients [57]. HEPs can form a highly dynamic equilibrium and can be considered together to describe the cerebral bioenergetic state. The in vivo concentrations of AMP and ADP are usually below the detection limits of ^31^P-MRSI. In addition, the reliable quantification of AMP/ADP is hindered by the spectral overlap to other metabolites [56]. The in vivo study of HEPs has already been applied to patients with PD [58], APD [59], AD [57], and HD [60], among others. In addition, NAD can also be quantified by ^31^P-MRSI [61,62]. NAD is an essential coenzyme in various redox reactions and can be present in a reduced (NADH) or oxidized (NAD^+^) state. NADH is the substrate of complex I of the ETC (the starting point of OXPHOS). In addition, experimental evidence suggests that the NADH/NAD^+^ ratio plays a crucial role in regulating OXPHOS and maintaining overall mitochondrial homeostasis [31]. The NADH/NAD^+^ ratio could be a surrogate marker of complex I activity and the general neuronal bioenergetic and redox state. Based on the substantial spectral overlap of NADH and NAD^+^, the ^31^P-MRSI-based differentiation of these metabolites requires elaborate experimental procedures. State-of-the-art ^31^P-MRSI platforms have proven the technical feasibility of NAD^+^/NADH differentiation in vivo [61,62]. Numerous studies have demonstrated the involvement of NAD metabolism in NDs, such as in AD or PD. Decreased NAD levels have been observed in both patient groups [31]. These findings finally led to the clinical evaluation of NAD precursors as potential treatments, e.g., for PD [63]. In general, MRSI studies benefit from high or ultrahigh magnetic field strengths to, e.g., shorten T1 relaxation times. Moreover, increased spectral resolution can substantially improve the differentiation of neighboring or partially overlapping peaks in ^31^P-MRSI-derived spectra [55].

### 2.2. Dynamic Measurements of OXPHOS Reaction Kinetics by ^31^phosphorus Magnetization Transfer Magnetic Resonance Spectroscopy Imaging

Combining ^31^P-MRSI with magnetization transfer preparation pulses (^31^P-MT-MRSI) can quantify OXPHOS-related kinetic reaction parameters of in vivo ATP metabolism [64]. In particular, three reaction rates have been the subject of previous research: iP ⇋ ATP, iP ⇋ PCr, and PCr ⇋ ATP [64]. Determining these steady-state reaction rates opens up many exciting opportunities to study brain energy metabolism in NDs in-depth. The ^31^P-MT-MRSI methodology has already been used to study patients with HD [60]. After visual stimulation, an increase in iP ⇋ PCr and iP ⇋ ATP ratios was observed in the occipital lobe of these patients. Following a similar experimental procedure, the investigational nutritional supplement *triheptanoin* restored the iP ⇋ PCr ratio in early-stage HD patients by providing TCA cycle substrates [65]. ^31^P-MT-MRSI can therefore be of particular interest to foster our understanding of impaired PCr metabolism, which has previously been implicated in HD [66]. However, the technical complexity of ^31^P-MT-MRSI hinders its widespread applicability in NDs research. ^31^P-MT-MRSI remains an active research area but has already been applied in other NDs, such as monogenic and idiopathic PD [58]. 

### 2.3. Quantitative Assessment of Mitochondrial Complex I by Positron Emission Tomography Imaging Radiotracer

Based on the prominent role of complex I disturbances in NDs, several PET radiotracers have been evaluated for quantitative imaging [67]. Out of these, the radiotracer ^18^F-BCPP-EF has entered clinical evaluation. Rodent studies have shown that ^18^F-BCPP-EF yields a high molecular specificity and undergoes dynamic changes following the administration of complex I inhibitors [68]. Studies in non-human primates additionally validated these findings in conscious, untreated, and MPTP-treated animals [69,70]. ^18^F-BCPP-EF is currently under investigation for the assessment of patients with NDs. In PD patients, decreased binding levels of ^18^F-BCPP-EF have been observed in several neuroanatomical key structures without reaching statistical significance [71]. In the same study, the longitudinal assessment of PD patients showed a trend without reaching the significance level [71]. In a multimodal PET imaging study in patients with AD, ^18^F-BCPP-EF binding is closely associated with the in vivo tau deposition but not the overall amyloid load [72]. ^18^F-BCPP-EF yields immense promises for evaluating mitochondrial dysfunction in patients with NDs. Even though more extensive studies are needed to assess the clinical applicability, a recent study has shown a convincingly high test-retest variability as an essential prerequisite for upcoming clinical trials [73].

### 2.4. Broadband Near-Infrared Spectroscopy to Dynamically Map Cytochrome c Oxidase Activity

NIRS helps to understand in vivo brain physiology and disease states non-invasively. Near-infrared light can, to a certain extent, penetrate living tissues. Here, absorbent molecules (so-called chromophores) can be measured by their respective tissue-related light attenuation [47]. Neuroscientists have widely used NIRS to measure changes in oxygenated (HbO_2_) and deoxygenated (Hb) hemoglobin, mainly following neuronal activation. Here, distinct frequencies of the near-infrared spectrum are usually used to map HbO_2_/Hb [47]. The mitochondrial enzyme *cytochrome c oxidase* (CCO) is one of the most abundant enzymes found in mammals. CCO can also serve as a chromophore in NIRS studies. Interestingly, the extinction spectrum of CCO changes based on its respective redox state (CCO vs. oxCCO) [74]. This phenomenon is of particular interest to measure OXPHOS disturbances by this methodology. However, lower CCO concentrations compared to HbO_2_/Hb chromophores pose a significant challenge for standard NIRS techniques [74]. CCO shows a broad absorption peak that is significantly different from HbO_2_/Hb chromophores [47]. Specific hardware setups are needed to disentangle the signals derived from the other chromophores in vivo. Ongoing advancements in standard NIRS technology led to the development of a method called *broadband NIRS* (bNIRS) [74]. bNIRS emits light within a wide frequency-range of the near-infrared spectrum to overcome these challenges [74]. Several in vitro and in vivo studies have demonstrated the successful separation of the CCO/oxCCO signal from HbO_2_/Hb changes by bNIRS [47]. Usually, physiological challenges are required to induce CCO/oxCCO signal changes. Within the given experimental setup, physiological parameters (e.g., the breathing rate) must be carefully monitored and considered as potential cofounders of derived findings [75,76]. The overall interpretation of derived CCO/oxCCO signal changes can be assisted by physiological models already available [77]. Unfortunately, these devices are currently not commercially available. In addition, many experimental options are available, including different algorithms for data processing, the precise definition of chromophore absorption spectra, and the number and variety of emitted wavelengths. 

## 3. Neuroimaging Assessments of In Vivo Oxygen Consumption

Mitochondrial dysfunction can lead to impaired brain oxygen metabolism [49]. In recent years, various neuroimaging techniques such as PET, MRI, and MRSI have been developed to study brain oxygen metabolism in vivo. These methods are discussed below, and we will stress how these methods can enhance our pathophysiological understanding of impaired oxygen metabolism in mitochondrial dysfunction. 

### 3.1. ^15^Oxygen-Positron Emission Tomography Imaging of In Vivo Oxygen Metabolism

The first method developed to probe brain oxygen metabolism was PET imaging with ^15^O_2_ tracers. Here, the patient inhales ^15^O_2_-enriched breathing gas, which enters the human body via the respiratory system, diffuses into the bloodstream, and reaches the brain tissue via the capillary bed. This process can be modeled by the single-tissue kinetic model [78,79]. Different kinetic parameters can be derived from this model; the extraction of ^15^O_2_ from the bloodstream into the brain tissue has been termed oxygen extraction fraction (OEF) [80]. Aerobic metabolic processes in the brain tissue then convert ^15^O_2_ to metabolic ^15^O-containing water (H_2_^15^O), which can recirculate through the body. To enable complete quantification of oxygen consumption, three-step methods consisting of three separate PET measurements on ^15^O_2_, C^15^O, and H_2_^15^O or C^15^O_2_ are usually used [80].

Nevertheless, there are already promising investigations regarding faster methods that require fewer PET scans using dual tracers [81]. In addition, arterial blood sampling has been evaluated to provide input variables for the ^15^O_2_-PET tracer kinetics model and the post-processing of neuroimaging data [78,82]. In general, blood sampling has been performed manually, but methodological advancements led to the automation of sampling regimes [81]. However, repetitive blood sampling is cumbersome and a significant burden for patients. The administration of inhalable radiotracers can be classified as continuous or bolus inhalation. The latter is characterized by short inhalation times of only up to one minute [79,83]. Bolus inhalation requires lower radiation doses and overall shorter scan times. The dynamic nature of the bolus inhalation regime helps to disentangle the temporal resolution of key aspects of oxygen metabolism, including the characterization of the cerebral blood flow (CBF) and the cerebral metabolic rate of oxygen consumption (CMRO_2_) [80]. Accordingly, the investigator must carefully select the advantages and disadvantages of respective experimental designs. Another challenge is the brain physiology itself; physiological parameters (such as heart rate, cardiac output volume, and breathing frequency, among others) should be kept as constant as possible. However, this crucial aspect is often challenged by the long and complex image acquisition procedures [80,84]. One significant limitation of ^15^O-PET studies is their complex logistical requirements [85]. For example, the tracer activity and flow rate must be set as accurately as possible; otherwise, large measurement errors can occur [86,87]. Since the 15O-tracer has a short half-life of approximately two minutes, a cyclotron should be as close as possible to the recording site, which substantially hinders the widespread clinical applicability [85,88]. In addition, handling gaseous radioactive tracers raises safety concerns that need to be addressed. Shielding devices and gamma camera-controlled exhaust systems for the scanner room have been successfully employed in the past [89].

The applicability of ^15^O_2_-PET within NDs research is critically dependent on the standardization of the experimental setup, image acquisition parameters, and reliable pre-and post-processing algorithms for subsequent neuroimaging analyses. Even though the ^15^O_2_-PET methodology has been a subject of interest for decades, extensive research is still needed to enter systematic clinical evaluation. Currently, ^15^O_2_-PET imaging is considered the gold standard for investigating cerebral oxygen metabolism, particularly by determining physiological key parameters such as the OEF or CMRO_2_ [80,90]. Therefore, although ^15^O_2_-PET imaging has been performed several times in humans, it has usually been done in healthy individuals and rarely in patients with NDs [91,92,93]. For further in-depth information about the historical development of the ^15^O_2_-PET methodology [82], relevant aspects for the experimental implementation of ^15^O_2_-PET, respective implications for the translation of ^15^O_2_-PET into clinical practice [94], as well as technical considerations [80], we kindly refer the interested reader to abovementioned review articles.

### 3.2. ^17^Oxygen-Magnetic Resonance Spectroscopy Imaging to Assess the Cerebral Metabolic Rate of Oxygen Consumption

At the end of the 1980s, an MRI-based alternative to ^15^O_2_-PET imaging emerged [90]. In contrast to the radioactive ^15^O isotope, the non-radioactive oxygen isotope (^17^O) has been identified as an NMR-active, non-zero spin system [95]. Similar to ^15^O_2_-PET imaging, ^17^O_2_-enriched gases are inhaled and diffused via the respiratory and vascular systems throughout the whole body of an organism. Regarding the assessment of mitochondrial dysfunction, H_2_^17^O is synthesized by complex IV during OXPHOS [38,88]. Preferably, ultrahigh-field detection of H_2_^17^O turnover by ^17^O-MRSI yields promising opportunities for the non-invasive assessment of cellular metabolism [96,97,98,99]. In this regard, the signal obtained by one measurement simultaneously reflects three processes occurring in vivo:oxygen consumption in the mitochondria via complex IV,leaching of H_2_^17^O from the brain via blood perfusion, andrecirculation of H_2_^17^O-containing blood. Here, H_2_^17^O is generated throughout the whole body of a given organism and returns to the brain via the bloodstream [38].

In ^17^O-MRSI neuroimaging studies, a high rate of H_2_^17^O generation indicates a high degree of oxygen consumption. Dynamic measurements of the oxygen consumption allow for the estimation of both CMRO_2_ and CBF [88]. To date, the quantification of brain CMRO_2_ via ^17^O-MRSI has been performed in animal models [95,100,101] and humans [102]. In addition, previous ^17^O-MRSI studies were able to differentiate CMRO_2_ in cerebral gray and white matter [102,103,104]. To a certain degree, in vivo ^17^O-MRSI allows for the spatially resolved detection of the NMR-signals, which is often considered an intrinsic limitation of heteronuclear MRSI studies [90,105]. However, specific MRI hardware setups and acquisition protocols are necessary for ^17^O-MRSI studies, in particular because of the short T1 and T2 relaxation times [88]. Compared to ^1^H- or ^31^P-MRSI, ^17^O-MRSI depends more on the use of high or ultra-high static magnetic field strength (usually ≥7 T) [96]. Spatial resolutions down to only a few millimeters have been obtained in animal studies. The feasibility of the dynamic in vivo ^17^O-MRSI method has already been confirmed for various magnetic field strengths ranging from 3 T to 16.4 T [106,107,108]. Likewise, feasibility studies on the brains of healthy and diseased individuals have yielded encouraging results [103]. Distinct hardware requirements, the limited availability of ^17^O-MRSI sequences, and the high production costs of ^17^O-enriched gases currently hinder widespread utilization in human-related studies. Several approaches are being evaluated to tackle these challenges, e.g., by reusing ^17^O-enriched gases [103].

### 3.3. Estimating Brain Oxygen Metabolism by Conventional Magnetic Resonance Imaging Methods

In contrast to the abovementioned MRSI-based approaches, methodological advancements of conventional MRI methods have also been adapted to study brain oxygen metabolism. These methods are based on classical ^1^H-MRI and can only indirectly offer insights into cerebral oxygen consumption using physiological models [109,110,111,112,113,114]. There are, in general, three indirect methods to map oxygen metabolism by conventional MRI methods [88]: 1.BOLD functional magnetic resonance imaging (fMRI). Here, physiological challenges (using interleaved hypercapnic and hyperoxic states) are demanded to calibrate the fMRI signal [115].2.T2-relaxation-under-tagging (TRUST) and susceptibility-based oximetry (SBO) [116,117]. Here, venous oxygenation is investigated using the spin labeling principle (similar to arterial spin labeling, ASL).3.Quantitative BOLD imaging to determine the cerebral venous blood volume and deoxyhemoglobin concentrations from transverse relaxation times (T2 or T2*) [118,119].

One significant advantage of these approaches is that clinically widely available MRI scanners can be used [120]. In general, alterations of the BOLD signal are composed of changes in blood flow and the change in oxygen metabolism following neuronal demand [120]. The so-called calibrated fMRI can estimate the latter. Calibration steps are performed beforehand to disentangle the different origins of the BOLD signal. For example, the study participant inhales predefined breathing gas mixtures, either containing higher ratios of CO_2_ or O_2,_ to alter the cerebral blood flow in a standardized and predictable manner. Respective changes of the BOLD signal in response to these gas mixtures can be combined with other physiological parameters (e.g., the breathing rate) as input variables for the model-based estimation of key aspects of the cerebral oxygen metabolism [120]. This methodology has evolved to the more advanced *dual-calibrated fMRI method*. Here, ASL and BOLD imaging have been combined to objectively quantify the changes in cerebral perfusion following respiratory challenges [35,121]. In the previous method, cerebral perfusion changes have only been assumed based on the known regulation of the neurovascular tone by altered CO_2_- or O_2_ levels. However, individual differences in cerebral perfusion following the standardized administration of predefined breathing gases have been black-boxed. Therefore, the combination of ASL and BOLD imaging in dual-calibrated fMRI studies adds another level of certainty to quantifying OEF, CMRO_2_, and CBF in vivo [121]. In summary, several promising methods are available to map cerebral oxygen metabolism, but most of them have yet to be established in comprehensive studies. In particular, multimodal neuroimaging assessments of cerebral oxygen metabolism will likely provide rewarding impulses for future research. 

## 4. Neuroimaging-Based In Vivo Assessment of Oxidative Stress

Oxidative stress is a well-established mechanism by which degeneration occurs in NDs [122]. In general, oxidative stress is characterized by an imbalance of pro-oxidant substances (e.g., hydrogen peroxide, H_2_O_2_) and cellular coping mechanisms (e.g., the enzymatic degradation of pro-oxidant molecules). The balance between pro- and antioxidant mechanisms is highly regulated. An excessive amount of pro-oxidant molecules can substantially damage various cellular components, including proteins, lipids, and DNA. However, many pro-oxidant molecules result from physiological processes and have important intracellular signaling functions [123,124]. During OXPHOS, by-products of ATP synthesis, e.g., ROS, can promote neuronal development and homeostasis if they do not exceed critical thresholds [123,125,126]. A plethora of cellular coping mechanisms against oxidative stress has been described. Two main concepts can be distinguished: 1.antioxidant enzyme systems (such as glutathione reductase, glutathione peroxidase, and catalases, among others) [127] and2.low-molecular-weight antioxidants (glutathione (GSH), uric acid, ascorbic acid, and melatonin, among others) [128,129].

However, suppose these protective mechanisms are disturbed or overloaded. In that case, excessive amounts of ROS can cause undesirable oxidative damage to lipids, proteins, and (mitochondrial) DNA, leading to cell degeneration and functional impairment through aberrant cell signaling and dysfunctional redox control [123,125]. The subsequent disturbance of cellular homeostasis can lead to a vicious cycle, i.e., if altered proteins become pro-oxidant substances [125]. Different aspects of human brain physiology make it susceptible to oxidative stress, e.g., by the high metabolic demand of neuronal tissue [124,126,130,131]. Mitochondria are prone to oxidative stress, mainly driven by the proximity of the ETC (as the primary generator of pro-oxidant molecules via OXPHOS) to crucial mitochondrial structures, such as the mitochondrial DNA (mtDNA). mtDNA damage caused by oxidative stress has already been identified as a frequent phenomenon in human aging [132,133,134]. Levels of oxidative stress exceeding physiological aging are a pathophysiological hallmark in many NDs [124,130,135]. In preclinical studies, oxidative stress can be measured with various methods. For example, paramagnetic electron resonance, ex vivo fluorescence staining, or high-performance liquid chromatography analyses have been proven helpful in elucidating the role of oxidative stress in NDs [136,137,138]. However, the development of neuroimaging methods for non-invasive, in vivo, and real-time assessments of oxidative stress often lacks clinical validation [139]. In general, the below-listed neuroimaging approaches can either assess one single molecular aspect of oxidative stress (i.e., by quantifying a single low-molecular-weight antioxidant) or by estimating the redox state of the human brain more holistically (i.e., by QUEnch-assiSTed-MRI, QUEST-MRI). These methods have intrinsic shortcomings, which may hinder their applicability in a clinical setting, considering the molecular complexity of oxidative stress and its respective regulation in vivo.

### 4.1. Proton-Magnetic Resonance Spectroscopy Imaging to Quantify Low-Molecular-Weight Antioxidants

Endogenous levels of ROS scavengers such as GSH and ascorbic acid have already been measured by ^1^H-MRSI in rodent models and humans [140,141,142]. In the following paragraph, we focus on GSH based on the eminent pathophysiological role and potential therapeutic implications [143]. Decreased GSH levels and the resulting vulnerability to oxidative stress have been implicated in many NDs [144,145,146]. Due to its thiol group, GSH has strong antioxidant properties and is therefore considered an important oxidative stress coping mechanism in neurons [147]. Reduced GSH can transfer electrons to ROS molecules and renders them harmless. Two oxidized GSH molecules can form a disulfide bond resulting in the synthesis of GSH disulfide (GSSG). The enzyme GSH reductase can reconvert the GSSG dimers into the biologically active, reduced GSH monomers. Thus, sufficient GSH levels are crucial to protect neurons against ROS-mediated cellular damage, support oxidative stress defense mechanisms, and maintain the physiological redox state in neurons [148,149,150]. Concludingly, the supplementation of GSH (or its precursor N-acetylcysteine, NAC) has been considered a dietary supplement in healthy individuals and a viable treatment regime in patients with NDs [151,152,153,154,155]. However, GSH/NAC supplementation may benefit only those individuals with low GSH levels before treatment, as has already been demonstrated in healthy individuals [156]. Significant challenges, such as the overall sensitivity of ^1^H-MRSI to detect GSH or the spectral overlap with other ^1^H-NMR active metabolites, necessitated technological advancements in the conventional ^1^H-MRSI methodology (e.g., by spectral editing techniques). For an extensive methodological review of these techniques, we refer to Choi et al. [157]. To date, the MEscher GArwood-Point RESolved Spectroscopy technique has been preferred in the previous studies [142,152,158]. The in vivo detection of cerebral GSH levels via ^1^H-MRSI will enhance our pharmacokinetic understanding of GSH supplementation and can advance personalized treatment decisions [152,158,159]. 

### 4.2. Probing Oxidative Stress by Over-Reductive Tissue State-Specific Radiotracer

The PET radioligand ^62^Cu-ATSM has been evaluated as a marker of oxidative stress in vitro and in vivo [160]. ^62^Cu-ATSM accumulates in brain regions with an intracellular over-reductive state, an epiphenomenon caused by oxidative stress [160]. In vitro studies have shown that higher intracellular retention of ^62^Cu-ATSM was present in cell lines with increased ROS due to mitochondrial dysfunction [161,162]. In PD patients, a correlation between increased ^62^Cu-ATSM accumulation in the striatum and the overall disease severity has been demonstrated [163]. Similar findings have also been shown for MND patients, where the increased ^62^Cu-ATSM retention in the motor cortex correlated with the functional disability [164]. Based on these promising findings, further studies will likely provide exciting insights into the disease mechanisms of other NDs. 

### 4.3. Iron-Sensitive Magnetic Resonance Imaging to Unravel the Generation of Oxidative Stress

Susceptibility-weighted imaging (SWI) is based on the measurement of local field inhomogeneities introduced by the presence of non-heme iron or copper [165,166]. The Fenton reaction (the oxidation of organic substrates catalyzed by iron salts with H_2_O_2_) has been identified as a primary source of ROS in NDs [167]. However, conventional SWI cannot be applied to differentiate between Fe^2+^ and Fe^3+^ ions, a crucial methodological hurdle to gaining deepened insights into the Fenton reaction-mediated promotion of oxidative stress. So far, only one phantom study has been published where the differentiation of Fe^2+^ and Fe^3+^ ions has been achieved by combining SWI and MRI relaxometry [168]. In vivo studies are still lacking but could substantially benefit NDs research.

### 4.4. Assessments of the Short-Term Response to Antioxidant Treatments by QUEnch-assiSTed MRI

Most of the previously discussed neuroimaging methods can give static insights into oxidative stress and may change within the individual disease course of patients with NDs. However, these methods are unlikely to recapitulate short-term effects or imminent changes in oxidative stress. The underlying idea that antioxidants could provide disease-modifying properties has been around for decades [8]. The design of clinical trials to address the unmet need for disease-modifying therapies in NDs encounters unique logistical challenges (e.g., caused by prolonged interventional periods to verify any disease-modifying effect). In addition, the multi-faceted regulation of oxidative stress in vivo requires careful decisions on promising candidate drugs, respective dosing regimes, and the correct timing of interventions within an individual disease course [169]. To overcome these hurdles, the neuroimaging-guided decision support and the time- and cost-effective evaluation of potential drug candidates is highly desirable. A recent modality called QUEST-MRI addresses many of the concerns above; here, the amount of paramagnetic ROS molecules can be indirectly measured via treatment-induced alterations of the in vivo relaxation rate (R1, 1/T1) [139]. QUEST-MRI does not require the administration of exogenous contrast agents, offers a high spatial resolution, and requires only standard MRI hardware [139]. This method can potentially broaden our understanding of the regional specificity of oxidative stress, can help evaluate novel antioxidant treatments in animal models or patients with NDs, and may guide personalized treatment decisions in the future. QUEST-MRI has already been successfully applied in rodent models and humans [170,171]. However, studies on patients with NDs are currently lacking. 

Overall, several neuroimaging methods offer promising opportunities for studying oxidative stress. However, necessary steps towards these methods’ clinical applicability must be taken. To date, bright and innovative approaches are available, but need procedural standardization and further experimental validation. The pre-experimental definition of viable neuroimaging markers must follow the proposed method of action of planned therapeutic interventions. Due to the complexity of oxidative stress in NDs, there will most likely be no one-size-fits-all approach. 

## 5. Summary

Mitochondrial dysfunction, in particular impaired OXPHOS and increased oxidative stress, plays a vital role in the pathophysiology of NDs. However, the clinical evaluation of mitochondria-targeted therapies in these disorders yielded conflicting results, and disease-modifying properties have not yet been reported. The complexity of in vivo mitochondrial dysfunction is a significant challenge for identifying viable treatment targets. The neuroimaging methods reviewed here can serve as feasible surrogate markers of mitochondrial dysfunction in vivo. Reliable biomarkers of mitochondrial dysfunction are urgently needed to improve the efficacy of clinical trials and guide personalized treatments in patients with NDs.

Furthermore, our current understanding of mitochondrial dysfunction in NDs appears to be too simplistic. Distinct facets of in vivo OXPHOS and oxidative stress, as well as their neuroanatomical, cellular, and subcellular compartmentalization, need to be evaluated within the scope of existing experimental neuroimaging techniques. Findings from distinct spatial and temporal practical scales are required in order to ensure their clinical applicability. The reviewed methods offer unique strengths and shortcomings for human use. These extend but are not limited to the invasiveness of a procedure, human safety concerns (e.g., from radiation), logistical hurdles (e.g., the required hardware), the overall acquisition time, the varying spatial and temporal resolution, the reproducibility of imaging results, the detection sensitivity, acquisition times, and the pathophysiological specificity of derived imaging contrasts. The standardization and multi-center evaluation of these methods are necessary prerequisites for human-related studies and their translation into clinical practice. The ongoing methodological advancements will foster a deepened understanding of mitochondrial dysfunction in NDs and probably have implications for the development of targeted therapies. 

## Figures and Tables

**Figure 1 ijms-23-07263-f001:**
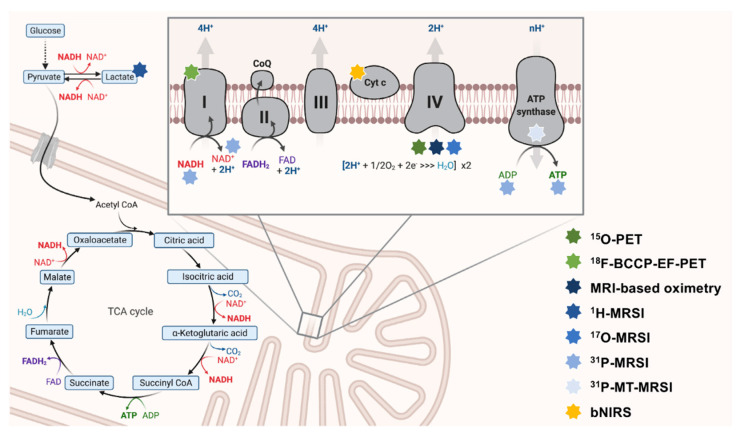
The interconnectedness of mitochondrial metabolism and OXPHOS-targeted neuroimaging approaches. Here, key aspects of OXPHOS are depicted. The ETC is schematically magnified (right upper corner). Multi-colored stars (legend: right-lower corner) indicate the respective neuroimaging modalities employed to map this particular aspect of metabolism. ^18^F-BCCP-EF: 2-tert-butyl-4-chloro-5-(6-(2-(2(^18^F)fluoroethoxy)-ethoxy]-pyridine-3-ylmethoxy)-2H-pyridazine-3-one. ADP: adenosine diphosphate. ATP: adenosine triphosphate. bNIRS: broadband near-infrared spectroscopy imaging. CoQ: coenzyme Q10. Cyt c: cytochrome c. ETC: electron transport chain. FAD/FADH^2^: flavin adenine dinucleotide. MRSI: magnetic resonance spectroscopy imaging. MT-MRSI: magnetization transfer magnetic resonance spectroscopy imaging. NAD^+^/NADH: nicotinamide adenine dinucleotide. OXPHOS: oxidative phosphorylation. PET: positron emission tomography. TCA: tricarboxylic acid.

**Figure 2 ijms-23-07263-f002:**
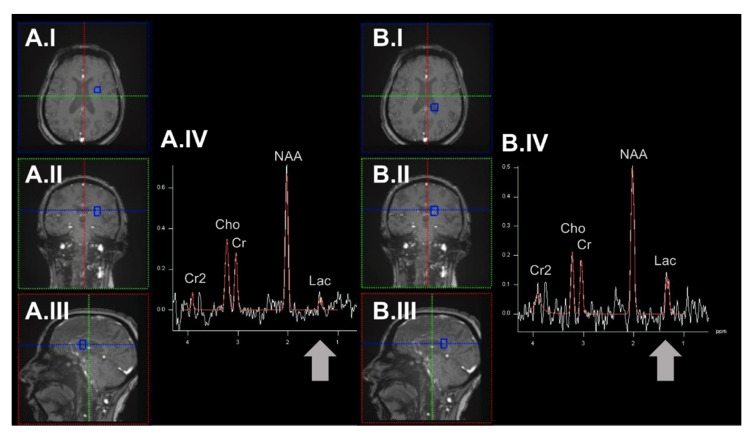
Exemplary determination of region-specific lactate levels by employing the ^1^H-MRSI methodology. Here, we demonstrate spectra derived from an example voxel (highlighted in blue) in the brain parenchyma of the left hemisphere (panel (**A**)) and the posterior portion of the left lateral ventricle (panel (**B**)). The respective voxel placement is shown in axial (**A.I/B.I**), coronal (**A.II/B.II**), and sagittal orientation (**A.III/B.III**). In this example, higher lactate levels can be observed in the CSF of the lateral ventricle. ^1^H-MRSI: proton magnetic resonance spectroscopy imaging. Cho: choline. Cr/Cr2: creatinine. CSF: cerebrospinal fluid. Lac: lactate. NAA: N-acetyl aspartate. ppm: parts per million.

**Figure 3 ijms-23-07263-f003:**
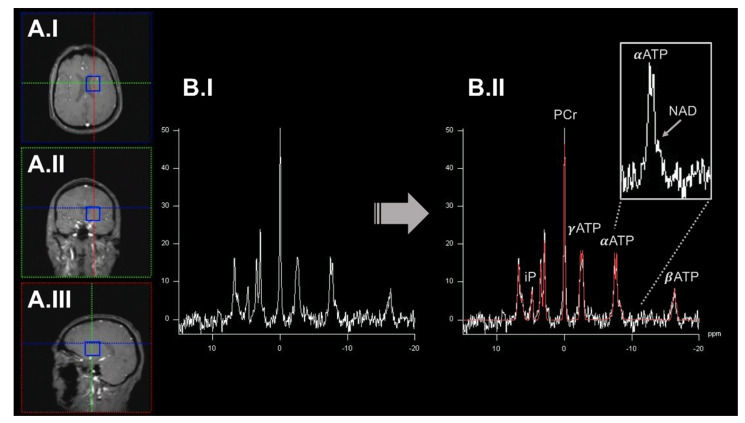
An exemplary illustration of ^31^P-MRSI acquisition workflow. Here, we highlighted an acquired spectrum from a single voxel (blue box) displayed in axial (panel (**A.I**)), coronal (panel (**A.II**)), and sagittal (panel (**A.III**)). The preprocessed (before peak-annotation) spectrum is shown in panel (**B.I**). In panel (**B.II**), the different peaks of the spectrum have been annotated based on a model fit (highlighted in red) by incorporating prior knowledge. Metabolite levels can be quantified by calculating the respective area under the peak. ATP gives rise to three signal peaks (α/β/γATP) based on the number of phosphorus nuclei in this molecule. Some metabolite peaks overlap (e.g., αATP and NAD; right upper corner). Here, elaborate experimental setups, acquisition protocols, and higher static magnetic field strengths are desirable to disentangle these metabolites. ^31^P-MRSI: ^31^phosphorus magnetic resonance spectroscopy imaging. ATP: adenosine triphosphate. iP: inorganic phosphate. NAD: nicotinamide adenine dinucleotide. PCr: phosphocreatinine. ppm: parts per million.

## Data Availability

Not applicable.

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
