# Peer review of "Neuroimaging Methods to Map In Vivo Changes of OXPHOS and Oxidative Stress in Neurodegenerative Disorders"

_ijms, 2022, doi:10.3390/ijms23137263_

Round 1
Reviewer 1 Report
This is a timely and very well written narative review on the topic. There really is little to add from my side. Maybe one thing: the authors rightfully point toward the complexity and heterogeneity of mitochondrial functions which may be affected in the path of neurodegenerative disorders. However, in some cases (eg. PD) we may have more specific information from genetic cases and pathway analyses from risk genes that could highlight particular elements. I was wondering if the authors deem worthwhile summarizing genetic information on mitochondrial pathways implicated in PD and other ND.
Author Response
We thank the reviewer for her/his helpful suggestions and comments. Based on the scope of the present manuscript we tried to restrict ourselves to neuroimaging methodologies to map mitochondrial dysfunction in patients with neurodegenerative disorders. We agree, however, with the reviewer that causative gene mutations and pathway analyses may be of interest to the readership of IJMS and would like to forward the interested reader to a review article that highlights these aspects in patients with Parkinson’s disease (as a neurodegenerative model disease). Please refer to the added paragraph in the introduction section:
“Genetic insights and pathway-based analyses help to elucidate the complexity of mitochondrial dysfunction in NDs. We refer the interested reader to a review article illustrating how these approaches help to identify potential treatment strategies in patients with genetic and non-genetic Parkinson’s disease as a neurodegenerative model disease (Prasuhn et al., 2021)”
Prasuhn, Jannik, and Norbert Brüggemann. "Gene Therapeutic Approaches for the Treatment of Mitochondrial Dysfunction in Parkinson’s Disease." Genes 12.11 (2021): 1840.
Reviewer 2 Report
In this review of "Probing mitochondrial dysfunction in patients with neurodegenerative diseases: A narrative review on neuroimaging based approaches", authors presented methodological spectrum includes positron emission tomography (PET), nuclear magnetic resonance (NMR), and near-infrared spectroscopy (NIRS), to study alterations in oxidative phosphorylation (OXPHOS) and oxidative stress, which deeply related to neurodegenerative diseases. It also discussed about the advantages and shortcomings of the different neuroimaging methods and their application for future human clinical research.
The following concerns should be addressed for further publishing consideration.
1. As you mentioned, "Whether mitochondrial dysfunction can be considered the primary driver or simply a bystander of neurodegeneration largely remains to be elucidated". Maybe it could be better to change the title into about neuroimaging methods to map in-vivo changes of OXPHOS and oxidative stress.
2. In line 18, the magnetic resonance and magnetic resonance spectroscopy belongs to the nuclear magnetic resonance (NMR) method as Section 1.5 described
3. Please double check and correct the typos in the manuscript, such as Section 3.1, the number of 15O isotope should be superscript; in line 608 section number is 4.4.
Author Response
We thank the reviewer for his nuanced comments and helpful suggestions.
The following concerns should be addressed for further publishing consideration.
- As you mentioned, "Whether mitochondrial dysfunction can be considered the primary driver or simply a bystander of neurodegeneration largely remains to be elucidated". Maybe it could be better to change the title into about neuroimaging methods to map in-vivo changes of OXPHOS and oxidative stress.
>> We agree with the reviewer that our title could reflect the focus of our manuscript more precisely and changed the title accordingly to:
“Neuroimaging methods to map in-vivo changes of OXPHOS and oxidative stress in neurodegenerative disorders”
- In line 18, the magnetic resonance and magnetic resonance spectroscopy belongs to the nuclear magnetic resonance (NMR) method as Section 1.5 described
>> We agree with the reviewer and made respective changes to the revised version of our manuscript.
- Please double check and correct the typos in the manuscript, such as Section 3.1, the number of 15O isotope should be superscript; in line 608 section number is 4.4.
>> The whole manuscript underwent in-depth proofreading addressing typos and formatting errors. Please refer to the highlighted changes in the revised manuscript.
Reviewer 3 Report
In this narrative review, the authors underlined several neuroimaging methods for probing mitochondrial dysfunction, providing a general overview of the current biological understanding of mitochondrial dysfunction in neurodegenerative disorders focusing on neuroimaging methods to map in vivo changes of OXPHOS and oxidative stress.
The topic is adequately addressed and discussed, I have no criticisms to raise for the manuscript which is well written providing an informative, detailed and critical overview on this topic of great interest and in my opinion the review can be accepted for publication in its current form.
Author Response
We thank the reviewer for the positive evaluation and the efforts in handling our manuscript.
Round 2
Reviewer 2 Report
The authors have done all the necessary correction and now the manuscript could be accepted in this current version.